# Functional Analysis of the Ribosomal uL6 Protein of *Saccharomyces cerevisiae*

**DOI:** 10.3390/cells8070718

**Published:** 2019-07-13

**Authors:** Lidia Borkiewicz, Mateusz Mołoń, Eliza Molestak, Przemysław Grela, Patrycja Horbowicz-Drożdżal, Leszek Wawiórka, Marek Tchórzewski

**Affiliations:** 1Department of Molecular Biology, Maria Curie-Skłodowska University, Akademicka 19, 20-033 Lublin, Poland; 2Department of Biochemistry and Molecular Biology, Medical University of Lublin, Chodźki 1, 20-093 Lublin, Poland; 3Department of Biochemistry and Cell Biology, University of Rzeszów, Zelwerowicza 4, 35-601 Rzeszów, Poland

**Keywords:** uL6, ribosome, GTP-ase associated center (GAC), ribosomal biogenesis, lifespan, aging, ribosomal gene duplication

## Abstract

The genome-wide duplication event observed in eukaryotes represents an interesting biological phenomenon, extending the biological capacity of the genome at the expense of the same genetic material. For example, most ribosomal proteins in *Saccharomyces cerevisiae* are encoded by a pair of paralogous genes. It is thought that gene duplication may contribute to heterogeneity of the translational machinery; however, the exact biological function of this event has not been clarified. In this study, we have investigated the functional impact of one of the duplicated ribosomal proteins, uL6, on the translational apparatus together with its consequences for aging of yeast cells. Our data show that uL6 is not required for cell survival, although lack of this protein decreases the rate of growth and inhibits budding. The uL6 protein is critical for the efficient assembly of the ribosome 60S subunit, and the two uL6 isoforms most likely serve the same function, playing an important role in the adaptation of translational machinery performance to the metabolic needs of the cell. The deletion of a single *uL6* gene significantly extends the lifespan but only in cells with a high metabolic rate. We conclude that the maintenance of two copies of the uL6 gene enables the cell to cope with the high demands for effective ribosome synthesis.

## 1. Introduction

The ribosome is a complex macromolecular machinery dedicated to protein synthesis. Both in prokaryotes and eukaryotes, ribosomes consist of two ribosomal subunits (r-subunits): Small and large, which can reversibly associate with each other forming a complete ribosome unit [1]. The structural organization of the ribosome facilitates cooperation of several active centers. The small r-subunit comprises a decoding center responsible for selection of an appropriate tRNA based on the mRNA codon [2]. The large r-subunit contains two centers: a peptidyltransferase center, where the formation of the peptide bond between a nascent polypeptide chain and a new amino acid occurs [2], and a GTPase-associated center (GAC) responsible for recruitment and stimulation of translational GTPases (trGTPases) whose activity accelerates the translation process [3,4]. The ribosomal subunits are composed of RNA molecules (rRNA) combined with ribosomal proteins (r-proteins) [5]. The rRNA component is responsible for the main events taking place on the ribosome, namely, decoding and peptidyltransferase activity, while r-proteins assist the ribosome in its functioning. R-proteins are critical for ribosome assembly as well as structure stabilization and provide regulatory potential for the translational machinery [2,6]. It has been postulated that r-proteins may also provide heterogeneity within the ribosomal pool, contributing to so-called specialized ribosomes, which may regulate the expression of genetic information at the translational level [7]. The eukaryotic ribosome, for example in yeast *Saccharomyces cerevisiae*, consists of four invariant rRNA and 78 proteins. Although 137 genes code for r-proteins, a major part of r-proteins (59) in yeast are encoded by a pair of genes, which could be regarded as paralogs [8,9]. The presence of two paralogous genes in one cell is a consequence of a genome-wide duplication event maintained in both lower and higher eukaryotes [10,11]. In most cases, proteins encoded by a pair of genes are identical or nearly identical [9]. Deletion of both duplicated genes often leads to synthetic lethality, which confirms that both proteins are needed for at least one essential function [12]. However, there are also data showing functional divergence between paralogs [13,14]. Until now, the reasons for the maintenance of two highly similar proteins are not fully understood. One of the possible explanations is partitioning of the ancestral gene functions (sub-functionalization). The second option is the development of a new function/-s (neo-functionalization) by one or both duplicated genes with simultaneous expense of the ancestral gene function. Finally, a combination of these events might occur as well [12,14,15,16]. Moreover, there are numerous examples providing evidence that r-proteins are a source of specialized ribosomes in *Escherichia coli*, yeast, *Dictyostelium*, *Arabidopsis*, zebrafish, and mice [17,18,19,20]. Thus, it can be claimed that r-proteins in the form of paralog pairs display at least partial redundancy, extending the regulatory potential of ribosomes. 

The translation process is intimately linked with the aging or longevity phenomenon, and perturbation within the translational machinery (especially within the r-protein composition) can extend the lifespan in yeast, worms, flies, and mammals [21]. Aging is a multifactorial process manifested through a decline in physiological functions, limiting biological processes, increasing the risk of disease, and ultimately leading to death. Although aging is a complex biological process shared by all living organisms, the molecular causes of aging are hardly understood. In general, it is thought that caloric restriction significantly improves the functioning and increases the lifespan of many model organisms [22], and because protein biosynthesis is the most energy consuming process in all living organisms, all perturbations within this pathway restrict energy consumption, and at the same time, may extend the lifespan. For instance, lifespan analyses have shown that a mutation of many genes associated with the ribosome, especially affecting the performance of the large ribosomal subunit, or depletion of other proteins aiding the translational machinery influences aging of evolutionarily disparate organisms. Depletion of r-proteins or translation factors can significantly slow aging [9,23,24,25,26,27,28]. However, it was shown that not all yeast strains with large ribosomal subunit protein deletion were long-lived [27]. Interestingly, the impact on aging was distinct even in the case of r-protein paralogs that share high identity in the amino acid sequence. In some paralog pairs, the effect of deleting one of the genes significantly increased the lifespan while deleting the other one did not, suggesting some functional divergence within the particular isoforms [27].

Here, we have undertaken functional analysis of one of the large ribosomal subunit proteins, uL6 (former name RPL9, L9) [29], which is located within the GAC. It was postulated that during ribosome biogenesis uL6 might be a key protein for the assembly of functional 50S subunits at the late stage in *E. coli* [30], but the role of the uL6 r-protein, especially its paralogs in eukaryotes, remains equivocal. Studies on *Drosophila melanogaster* and zebrafish showed the importance of uL6 for normal growth, development, and viability [31,32]. Moreover, the uL6 protein was suggested to be involved in the mouse mammary tumor virus (MMTV) particle assembly process [33]. The uL6 protein in yeast is encoded by two paralogous genes, *RPL9A* and *RPL9B* [34], hereafter named *uL6A* and *uL6B*, respectively, according to the r-protein nomenclature used in the study [29]. Here, we have also raised an issue of the duplication of the uL6 gene in the context of the functioning of the translational machinery together with its consequences for aging of yeast cells. 

## 2. Materials and Methods

### 2.1. Strains, Plasmid Construction and Cell Growth 

The yeast strains used in this study are listed in Table 1. All yeast strains are in the BY4741 genetic background. The single *Δul6A* or *Δul6B* mutant and wild type BY4741 strains were obtained from the (EUROSCARF, Oberursel, Germany) yeast strain and plasmid collection. The conditional GAL::uL6A *Δul6AΔul6B* mutant strain was constructed in *Δul6A* genetic background strain by genetic modification involving transformation of pYES2 plasmid born copy of *uL6A* gene under galactose promoter and simultaneous deletion of the *uL6B* gene replaced by LEU marker using homologous recombination technique [35]. The deletion mutant strains with plasmid born complementation of uL6A by uL6B and uL6B by uL6A were obtained by introducing plasmids carrying *uL6A* or *uL6B*, respectively, to the single *Δul6A* or *Δul6B* mutant strains. Plasmids for the complementation of uL6A and B were generated on a basis of a tetracycline-repressive pCM190 vector, using standard genetic techniques. Yeast were grown on YPD (1% yeast extract, 2% peptone, and 2% glucose) or YPGal (1% yeast extract, 2% peptone, and 2% galactose) medium at 30 °C, 200 rpm unless otherwise stated.

### 2.2. Protein Structure Prediction

All 3D structures were downloaded from the Protein Data Bank (PDB) and displayed with PyMOL (The PyMOL Molecular Graphics System, version 1.5.0.4, Schrödinger, LLC, New York, NY, USA). The alignment of amino acid sequences was performed with ClustalX1.8 with default options [36], and it was manually edited.

### 2.3. Yeast Polysome Profile Analysis

Polysome profile analyses were performed by centrifugation of total cell extracts in 7–47% linear sucrose gradients. Cells were grown to OD_600_ (A_600_) 0.4–0.6 in YPD, YPGal or appropriate minimal medium (SD) and treated with cycloheximide (CHX) to the final concentration 100 µg/mL for 20 min to stabilize the translating ribosomes on mRNA. Next, cells were harvested by centrifugation and resuspended in lysis buffer [10 mM Tris-HCl pH 7.5, 100 mM NaCl, 30 mM MgCl_2_, 100 μg/mL CHX, 1 mM PMSF, 6 mM β-mercaptoethanol, 1 nM pepstatin A, 10 nM leupeptin, 10 ng/mL Aprotinin, 200 ng/mL heparin, and RNase Inhibitor (Sigma-Aldrich, Saint-Louis, MO, USA)] and disrupted by vigorous shaking with glass beads at 4 °C. The cell lysate was precleared by centrifugation at 12,000 rpm, 10 min, 4 °C (Rotor 12154-H; SIGMA, Osterode am Harz, Germany). Aliquots of lysate equivalent to A_260_ 12 units were loaded on linear sucrose gradient, centrifuged for 4,5 h at 26,500 rpm and 4 °C in a SW32Ti rotor (Beckman–Coulter, Brea, CA, USA). The resulted fractions were analyzed using an ISCO Brandel Density Gradient Fractionator (Brandel Isc, Gaithersburg, MD, USA 2.4. Translational Fitness Determination by ^35^S-Radiolabeled Methionine Incorporation.

Cells were grown to OD_600_ 0.5–0.6, washed with deionized water and resuspended with methionine-depleted SD minimal medium (SD-Met). After 15 min of cell cultivation at 30 °C unlabeled-methionine was added to the final concentration 50 mM and 37 kBq of ^35^S-Methionine (37 TBq/mmol, Hartmann Analytic, GmbH, Braunschweig, Germany) was added at time 0 (T0). At 10-min intervals, the OD_600_ of the cultures was measured and 1 mL aliquots of the cultures were collected for protein precipitation with ice-cold 50% TCA (final concentration 10%). Next, proteins collected on Whatman GF/C filters were counted in a scintillation counter Beckman LS6000SE. The translation impairment was determined by comparison of the incorporation rate (cpm/OD_600_/min) of mutant cells with wild type cells, plotted as a function of time. The results were expressed as the mean percent of wild type activity.

### 2.4. Determination of Budding Lifespan

The budding lifespan of individual mother yeast cells was defined as the number of mitotic cycles (buddings) during the cell’s life. After overnight growth, cells were arrayed on a YPD, YPGal, or SD-Ura plate using a micromanipulator. The budding lifespan was determined microscopically by a routine procedure with the use of a micromanipulator as described previously [37]. The number of buds formed by each mother cell signifies its reproductive potential (budding lifespan). In each experiment, 45 single cells were analyzed. The results represent measurements for at least 90 cells analyzed in two independent experiments. The analysis was performed by micromanipulation using the Nikon Eclipse E200 optical microscope with an attached micromanipulator.

### 2.5. Determination of Total Lifespan

The total lifespan was defined as the length of life of a single mother cell expressed in units of time. The total lifespan was calculated as the sum of reproductive and post-reproductive lifespans. The reproductive lifespan was defined as the length of time between the first and the last budding and the post-reproductive lifespan as the length of time from the last budding until cell death. The lifespan of the *Saccharomyces cerevisiae* yeast was determined as previously described by [38] with small modification [37]. Ten microliter aliquots of an overnight grown culture of yeast were collected and transferred on YPD, YPGal, or SD-Ura plates with solid medium containing Phloxine B (10 μg/mL). Phloxine B was used to stain *Saccharomyces cerevisiae* dead cells. Dead yeast cells lost membrane integrity and Phloxine B entered cell space giving pink/red coloration of cytosol. In each experiment, 45 single cells were analyzed. During manipulation, the plates were kept at 28 °C for 15 h and at 4 °C during the night. The results represent measurements for at least 90 cells analyzed in two independent experiments. The analysis was performed by micromanipulation using the Nikon Eclipse E200 optical microscope with an attached micromanipulator.

### 2.6. Cell Budding Ability and Viability of GAL::uL6A Strain 

For verification of the cells budding, 20 µL of the cell suspensions were spotted on the plate with solid YPD or YPGal medium, and the pictures of the cells were taken using the Nikon Eclipse E200 microscope equipped with the Olympus DP26 digital camera at the beginning of the experiment and after 0 h, 24 h, and 48 h. 

For determining death cells, staining with PI was used. Cells were suspended in PBS and stained with 5 μg/mL propidiumiodide (Sigma-Aldrich, Saint-Louis, MO, USA) for 15 min in the dark at room temperature. Fluorescence pictures were taken with Olympus BX-51 microscope equipped with a DP-72 digital camera and cellSens Dimension software (Olympus, Tokyo, Japan). Dead cells were red fluorescent. 

### 2.7. Statistical Analysis

The results represent the mean ± SD values for all cells tested in two independent experiments. The differences between the wild type and the isogenic mutant strains were estimated using the one-way ANOVA and Dunnett’s post-hoc test. The values were considered significant if *p* < 0.05. Statistical analysis was performed using the Statistica 10 software (StatSoft Inc., Tulsa, OK, USA).

## 3. Results

### 3.1. Depletion of uL6 Isoforms is not Lethal for Yeast Cells

The uL6 protein is located in the ribosomal GAC (Figure 1A), i.e., the active center responsible for the interaction with trGTPases and participating in stimulation of GTP hydrolysis by translational factors [39]. As most of the yeast duplicated r-proteins, uL6A and uL6B share high homology in the amino acid sequence (98.5%) (Figure 1B). Both paralogs differ from each other in only four amino acids (Figure 1B). The substitutions are Val_12_/Ile, Gly_61_/Asp, Thr_188_/Val, and Leu_191_/Met. The Val/Ile and Leu/Met represent conservative substitutions, whereas Thr/Val and Gly/Asp cause significant changes, which may result in different charge distribution in the protein. However, Val/Ile, Thr/Val, Leu/Met residues are clustered at the exposed to the cytoplasm region, whereas Gly/Asp is buried in the interface, having a potential to interact with rRNA (Figure 1 A and C). Thus, uL6A containing a Gly residue at position 61 may interact with rRNA more firmly than uL6B containing an Asp residue at this position, which may destabilize protein-rRNA interactions. 

To investigate the functional impact of uL6 on yeast metabolism and translational apparatus, initially we used the individual yeast strains depleted of either *uL6A* or *uL6B* genes. The phenotype screening analysis of the yeast mutants exposed to different stress conditions (e.g. availability of diverse carbon sources, osmotic stress, high/low temperature stress, oxidative stress, proteotoxic stress, or factors causing cell wall disorder) did not reveal any significant differences in comparison to the wild type strain (data not shown). This information suggests that the A and B isoforms of uL6 protein do not have a significant role in adaptation of yeast cells to changing environmental conditions. Therefore, to investigate the uL6 function in detail, we constructed a conditional strain, GAL::uL6A, based on a *Δul6AΔul6B* mutant strain using a pYES2 vector. In this strain, the gene for the uL6A protein is ectopically expressed under a GAL1 promoter. The growth of the strain on galactose-supplemented solid medium was similar to the growth of the wild type strain (Figure 2A); however, the growth was totally abolished on medium supplemented with glucose (Figure 2B), suggesting that the uL6 protein is essential for yeast survival. However, growth analyses on liquid medium showed that, after the shift from the permissive galactose-based-medium to the repressive glucose-based-medium, the GAL::uL6A strain significantly slowed down the growth and did not show exponential growth as is the case for the wild-type strain (Figure 2C). The slow but noticeable growth of the mutant strain can be probably attributed to the stability of the ribosomal particles, which are present in the cell for more than 24 h and can support residual protein synthesis and at the same time slow growth. These data were confirmed by microscopic observations of the conditional mutant strain budding ability on solid YPD or YPGal medium and viability on YPD liquid medium (Figure 2D–F), and as was shown on Figure 2D–E, cell volume does not change during 24 h of incubation. GAL::uL6A cells grown on YPD medium after 24 h showed the capability of few cell cycle progressions. Interestingly, after 96 h of incubation in liquid YPD medium, only a few percent of GAL::uL6A cells were not viable (Figure 2F), which clearly shows that the ribosomal uL6 proteins are not essential for cell survival.

### 3.2. Depletion of uL6A or B Isoforms Reduces Global Translation

To get insight into the involvement of both uL6 paralogs in functioning of the translational machinery, we examined the global translation rate by measurement of radiolabeled ^35^S-methionine incorporation for single *Δul6A* and *Δul6B* mutants with respect to the wild type strain. The efficiency of global translation recorded using single deletion mutants was lower for both mutant strains compared to the wild type (Figure 3), indicating that lack of one isoform of the uL6 protein in a particular yeast strain has a negative effect on the translational machinery. However, the decrease rate is 20% in comparison to the wild type, supporting the process to an extent that is sufficient to maintain the growth of single deletion mutant strains in various environmental conditions. Moreover, there were no differences in the rate of the global translation efficiency observed between both single deletion mutants, suggesting that the maintenance of these two uL6 paralogs is not strictly connected with the efficiency of translation.

### 3.3. Lack of uL6A or uL6B Inhibits Ribosome Biogenesis

To cast more light on the role of uL6A and uL6B on the translational machinery, we performed polysome profile analysis of *Δul6A* and *Δul6B* mutant strains; this method allows characterization of the translational machinery through monitoring the engagement of particular ribosomal elements in the translational process. As shown in Figure 4A, single *Δul6A* and *Δul6B* mutants grown in YPD medium exhibited almost identical polysome profiles to wild type cells. Especially considering the P/M and 40/60 parameters, the performance of the translational apparatus of the single mutants resembles that found in the wild type strain. However, there is one pronounced difference, namely, so-called half-mers were observed in the polysome profiles of *Δul6A* and *Δul6B* mutant cells. The half-mers represent the 43S pre-initiation complex on the mRNA and may arise because of a defect in the initiation step of the translational cycle or perturbation in 60S biogenesis; in any case, the presence of half-mers indicates impairment in 40S and 60S subunit joining during initiation. Subsequently, the polysome profile was analyzed for the GAL::uL6A strain grown on glucose-based repressive medium for 12 h or 24 h. In the control experiment (yeast growth on galactose-medium), the GAL::uL6A mutant displayed no significant changes in the polysome profile in comparison to the wild type strain (data not shown). However, it should be pointed out that the polysome content was lower because galactose in the medium was not as efficiently used as glucose by the yeast cells, and consequently, the metabolism rate was lower. The switch to the glucose-based medium and repression of *uL6A* gene expression resulted in dramatic changes in the polysome profile. After 12 h of growth, remarkable reduction in the polysome profile was observed, accompanied by simultaneous accumulation of 40S and reduction of 60S subunits; these features were also well visible after 24 h of growth on glucose, with a further decrease in the polysomal fraction and, especially, disappearance of 60S (Figure 4B). The disappearance of the 60S subunit and 40S accumulation is a very pronounced feature, indicating that the presence of the uL6 protein is crucial for large ribosomal subunit biogenesis. 

### 3.4. Expression of uL6A or uL6B in Δul6A and Δul6B Mutants Restores the Translational Machinery Performance

The analyses of so-called translational fitness, including incorporation of radiolabeled ^35^S-methionine and especially polysome profile analyses using *Δul6A* and *Δul6B* mutants showed that lack of the A or B isoform impairs the translation efficiency and revealed the presence of prominent half-mers in the single deletion strains, irrespective of gene deletion. To test the influence of uL6A and uL6B excess on the translation and metabolism, we used single deletion mutant strains complemented with plasmids constructed to overexpress paralogous proteins in several configurations. For this purpose, we used a high-copy episomal plasmid pCM190 [40]. In the control experiment, the strains carrying the empty vector showed a polysome profile resembling that observed for the *Δul6A* and *Δul6B* mutants grown on rich medium with marked half-mers (Figure 5). Next, we tested whether one paralog can compensate the lack of another or whether an additive effect occurs. The complementation analysis showed that the additional uL6 protein copies (irrespective of the A or B isoform) restored the wild type phenotype in all analyzed cases, and we did not observe any abnormalities in the polysome profiles regarding the presence of the half-mers. In the case of the control experiment with the wild type strain, the expression of additional uL6 copies did not affect the polysome profile.

### 3.5. Deletion of uL6A or uL6B Affects the Budding Lifespan and the Total Lifespan of Yeast

As has been already reported [9,21,23,24,25,26,27,28], lowering the performance of the translational machinery may bring positive changes in the lifespan of eukaryotic cells, especially deletion of r-proteins from the 60S subunit. Thus, we checked whether deletion of single uL6 encoding genes or both paralogs affects the budding lifespan and survival. The budding lifespan (also called the replicative lifespan) is a measure of the number of daughter cells produced by the yeast mother cell, and although it is a standard used to determine the cell age, it actually determines its fertility rather than aging. In turn, the survival rate of the cells is determined by analyzing the total lifespan; it shows cell lifetime expressed in units of time and is a sum of the reproductive lifespan and post-reproductive time (from last budding to death). Thus, we analyzed the lifespan parameters applying two experimental set-ups, namely, with glucose- and galactose-based medium, using single- and double-deletion strains. Our data show that in the presence of glucose, lack of one of the *uL6A* or *uL6B* genes leads to a statistically significant increase in the mean budding lifespan (*p* < 0.001) up to 34 and 28 daughters in single deletion strains vs. 20 in the wild type strain. In turn, deletion of both genes drastically reduces the budding lifespan to 3 (*p* < 0.001) (Figure 6A). These differences were accompanied by a statistically significant increase in the reproductive lifespan of the single mutant strains and a reduced reproductive lifespan of the double mutant (*p* < 0.001) (Figure 6B). Interestingly, the total lifespan of the single mutants was slightly extended compared to wild type BY4741, but these were values not statistically significant (Figure 6D). In turn, the deletion of both versions of genes leads to more than a twofold increase in the lifespan (*p* < 0.001). The extension the life of the double mutant is associated with a significant extension of the post-reproductive time of life compared to the control (*p* < 0.001) (Figure 6C). 

On the other hand, the lifespan parameters of the analyzed yeast strains on galactose-based medium showed different results. The lack of one of the *uL6A* or *uL6B* genes did not change the budding lifespan in comparison to the wild-type strain, while deletion of both genes drastically reduced the budding lifespan, reaching 33 for the wild type vs. 17 for the double deletion strain (*p* < 0.001) (Figure 7A). Additionally, there were no changes in the reproduction time in the single deletion mutants, compared to the control, whereas the double mutant displayed a significantly reduced reproduction time (*p* < 0.001) (Figure 7B). Substantial differences were recorded in the case of the total lifespan and the post-reproductive time of life. In all cases, the mutant strains (single or double) had a significantly extended time, especially the double disruptant displayed a very long post-reproductive time having 104 h vs. 12 h in the wild type strain (Figure 7C,D). It should be underlined here that, in the yeast growth analysis on liquid and especially on the solid YPD medium (Figure 2B,C), we did not observe dynamic cell growth of the GAL::uL6A mutant. This shows that growth arrest is mainly associated with perturbation of cell cycle progression, which is manifested as a decrease in the replicative lifespan rather than cell death. However, in general, lack of one isoform of the uL6 protein exerts a significant impact on the lifespan parameters in the case of dynamically growing cells on glucose-based medium, contrary to conditions on galactose-based medium, where the impact is not very pronounced. 

## 4. Discussion

The yeast *Saccharomyces cerevisiae* genome contains ~530 pairs of paralogs/isoforms with highly similar sequences that evolved from a gene duplication event [41]. Interestingly, among 78 yeast r-proteins, 59 are duplicated and could be regarded as isoforms or paralog pairs [8,9]. Such protein pairs have been maintained through the evolution, showing the biological importance of gene duplication and indicating that individual isoforms may possess individually functional specificity [42]. It is thought that the presence of paralogs among ribosomal proteins may contribute to heterogeneity of the translational machinery. The pool of so-called specialized ribosomes having an altered structure due to the presence of a different subset of r-proteins or modifications to rRNAs may affect the expression of different subsets of mRNAs [20]. However, it should be underlined that there is no consensus on the contribution of each r-protein isoform to a specific cellular event.

In this study, we have focused our attention on the yeast uL6 r-protein, which is encoded by two *uL6A* and *uL6B* genes. The two isoforms of uL6 display high amino acid sequence similarity, having 98.5% identity. To cast light on the functional specificity of the uL6 paralogs, we have performed in vivo analyses involving yeast strains lacking particular genes encoding the A or B isoform of uL6. Yeast growth analyses of the *Δul6A* or *Δul6B* single yeast mutant strains did not reveal any significant differences between strains grown in various environmental conditions, suggesting that the function of either uL6A or uL6B is not directly connected with adaptation of yeast to changing environmental conditions. On the other hand, simultaneous depletion of both *uL6A* and *uL6B* genes caused synthetic lethality; the observation is in agreement with published data showing that uL6 belongs to proteins essential for yeast survival/growth [43]. Therefore, to study the uL6 function in vivo, we generated a conditional GAL::uL6A mutant strain, which allowed us to analyze the effect of lack of both copies of genes encoding the uL6 protein in cells in inducible conditions. In the GAL::uL6A strain, the expression of the uL6A paralog restored the growth of the double *Δul6A Δul6B* mutant strain in galactose-supplemented medium, while the growth of the conditional mutant in YPD solid medium was abolished. Interestingly, when the cells were transferred from permissive YPGal liquid medium to repressive YPD liquid medium, the GAL::uL6A strain displayed moderate growth. Considering the high stability of ribosomal particles, the initial growth of the GAL::uL6A mutant on glucose-based-medium was probably supported by the remaining ribosomal particles. As shown in Figure 4B, after 12 or even 24 h of growth in YPD medium 60S, 80S and residual polysomes are present, which may support slow growth. Moreover, the cell viability test showed that the yeast cells maintained full viability, indicating that the lack of uL6A and B forms is not required for cell survival, but the absence of uL6 proteins hampers only cell growth (Figure 2D–F). Thus, the question arises why the lack of the A and B forms of uL6 exerts a strong effect on yeast cell metabolism. In general, ribosomal proteins, being an integral part of the translational machinery, support ribosomes in protein synthesis and therefore depletion of many of them results in reduction of the translation rate [9,44,45]. However, our data show that individual isoforms of uL6 are not crucial for protein synthesis, as the depletion of either the *uL6A* or *uL6B* genes did not block global translation and only moderately reduced its rate by 20%; thus, both uL6A and uL6B paralogs support translation to a similar extent. Despite of the minor effect of uL6A and uL6B r-proteins in the quantitative aspect of protein synthesis (so-called translation fitness), the polysome profile analysis revealed that the uL6 protein is required for ribosomal biogenesis. In yeast cells, depletion of uL6A or uL6B or especially both proteins leads to a serious imbalance of 40S over 60S subunits and generation of half-mers, with simultaneous reduction of the polysomal fraction. The involvement of uL6 in the middle steps of pre-60S assembly during ribosomal biogenesis was suggested earlier in yeast on the basis of a large-scale proteomic approach [46], and the homologous bacterial protein was also reported to play a role in the assembly of the large ribosomal subunit [30]. The uL6 protein, being an integral part of the GAC element, represents an important ribosomal component building the ribosomal center responsible for the GTP-stimulatory effect of trGTPases. Based on structural analyses, the uL6 is bridging the sarcin-ricin loop (SRL) with the uL11 r-protein, and SRL-uL6-uL11 together form an interaction network and help to sensor the interaction of trGTPases with the ribosome [39]. Interestingly, the lack of eukaryotic uL11A/B is not as deleterious as the lack of uL6A/B (a double deletion strain has a slow growth phenotype [44]), although it is postulated that uL6 does not take part in direct stimulation of GTP hydrolysis by the GAC, contrary to uL11 [39,47]. It should be point out that uL6 is located close to the critical SRL and forms a broad web of interactions with rRNA [39] also in the case of bacterial counterpart [48]. Additionally, uL11, the partner of uL6, is taking part in cooperative binding of the uL10-L12 protein complex, the so-called ribosomal stalk, as it has been shown for bacteria [49]. Thus, it can be postulated that the ‘toxic’ effect of uL6 absence can be explained by the possibly important role of uL6 in formation/stabilization of the GAC, taking into account the SRL neighborhood and lack of uL6 may also hamper formation of the stalk on the 60S subunit. 

Our data cast more light on the open issue of the role of genome duplicate sequences in yeast. Although some paralogous r-proteins exhibited functional divergence [50], the phenotypic analyses of the single *Δul6A* or *Δul6B* yeast mutants did not reveal any differences between the functions of uL6A and uL6B, and it seems that both uL6 proteins share the same ancestral function. As proposed for duplicated genes encoding parts of macromolecular complexes, the functionally similar genes are preserved in one cell due to the different level, rate, dynamics or noisiness of its expression under selected conditions [51,52,53,54]. Indeed, the level of *uL6B* gene expression is tightly regulated; depending on the transcript copy number and expression of uL6B, it can be terminated by one of two pathways of transcription termination, namely by RNA polymerase II (Pol II)-associated cleavage and polyadenylation complex (CPF/CF complex), or Nrd1 complex. Moreover, the amount of uL6B mRNA can be increased in the absence of the uL6A transcript, indicating the existence of mechanisms of complementation between the paralogs [34]. These results corroborate our functional analyses, which show that maintenance of two uL6 copies enables the cell to adjust the translational machinery performance to metabolic needs. When one uL6 isoform is missing in the cell, symptoms of biogenesis defect, namely half-mers, are present, however, only when cells are cultivated on glucose-based medium, where the yeast cells have the highest growth rate. On the other hand, in the strain having only one gene encoding uL6 grown on galactose-based medium, no half-mers were detected, probably because the rate of metabolism in these conditions is slower and there are no high demands from the translational machinery to synthesize big amounts of proteins rapidly. This idea is supported by the complementation analysis, which showed that, irrespective of the genetic background (lack of A or B form of uL6), addition of extra-copies of uL6A or uL6B abolishes the defect in the translational machinery manifested in the form of half-mers. The observed phenomenon can be explained by the fact that the synthesis of new ribosomes composed of 80 proteins and 4 rRNAs is a fast, multiple-step pathway. To achieve this goal, maintenance of stoichiometrically precise balance in the expression of genes encoding r-proteins is critical for the cell, and we propose that, for some r-proteins that are critical for ribosome assembly, the genes remain duplicated to maintain the assembly homeostasis of the translational machinery, as is the case of uL6. Thus, we postulate that the functional duplication of the *uL6* gene is related to efficient production of uL6 proteins, which is critical for formation of the GAC on the ribosome. Our conclusion is also supported by analyses of yeast cell growth, performed at a single cell level, namely, measurement of the so-called lifespan. Many reports have described an influence of deletion of r-protein gene paralogs on the budding lifespan, especially r-proteins constituting the 60S subunit [27,55,56,57,58], indicating that perturbations within the translational machinery restrict energy consumption and at the same time may extend the lifespan [19]. We showed that the lifespan of single *Δul6A* and *Δul6B* mutants is variable and dependent on growth conditions. In general, the lifespan parameters were extended for single mutants grown on glucose-based medium, contrary to galactose-based medium, where we did not observe such discrepancies between the wild type and the single mutants as the yeast cell displayed a lower metabolic rate. Thus, the observation is in line with the general notion showing that reduction of the translational performance by deletion of a single *uL6* gene extends the lifespan in cells with a high metabolic rate. However, when the cell metabolism rate is low (growth on medium containing an unfavorable carbon source—galactose), the lack of one of the *uL6* genes is unnoticeable, probably because of the lower rate of 60S assembly and the lower demand for the uL6 protein. The observation is in full agreement with previous results, which showed that extension of the lifespan can be achieved by reduction of the metabolic rate as a consequence of lowering the temperature of growth [59] or as a result of mutations that significantly disturb protein biosynthesis [60]. Additionally, the single cell level approach showed that the double *Δul6AΔul6B* (GAL::uL6A) mutant strain has an extremely extended total lifespan, with concomitant reduction in the budding lifespan, explaining the observed phenomenon that such cells maintain full viability but are not able to produce daughter cells. 

In conclusion, the functional analysis of the A and B forms of the uL6 protein indicates that the two uL6 isoforms are functionally redundant, but the presence of duplicated *uL6A* and *uL6B* genes allows yeast cell to efficiently produce uL6 r-protein, which may meet the high requirements for 60S subunit assembly in the growth conditions when the cell displays a high metabolic rate. Thus, our data support the proposed hypothesis that partitioning of the ancestral gene functions, so-called sub-functionalization, occurs in the case of the uL6 protein, allowing yeast cells to cope with highly demanding metabolic conditions. 

## Figures and Tables

**Figure 1 cells-08-00718-f001:**
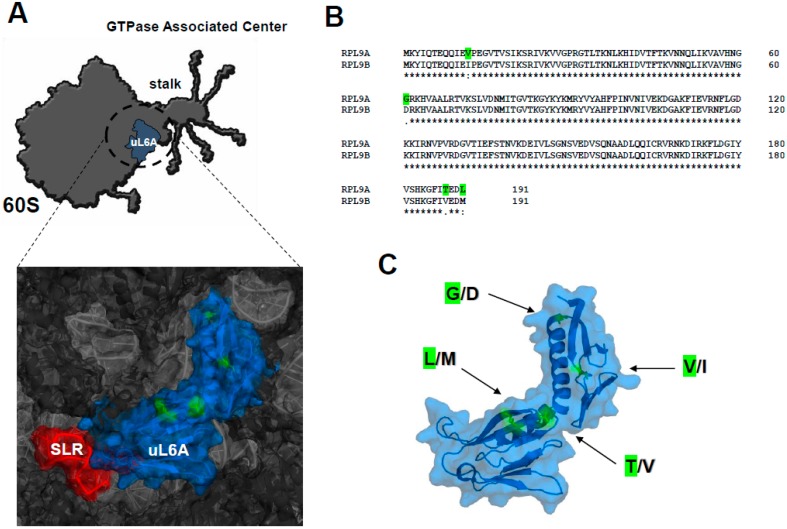
Structural overview of yeast ribosomal uL6A and uL6B proteins. (**A**) Model showing the position of the uL6A protein within the GTPase-associated center in the 60S subunit; *S. cerevisiae* 26 S rRNA (PDB code 3U5H) and proteins (PDB code 3U5I) from the 60S subunit are indicated as light gray and dark gray colors, respectively. The fitted schematic structure of the uL6A protein is marked in blue. The positions of mutated amino acids are marked in green. The sarcin-ricin loop (SRL) is marked in red. The stalk depicts the position of ribosomal P-proteins and uL10. (**B**) Amino acid sequence alignment of uL6A and uL6B proteins [36]. (**C**) Structural model of the uL6A protein with marked mutations positions.

**Figure 2 cells-08-00718-f002:**
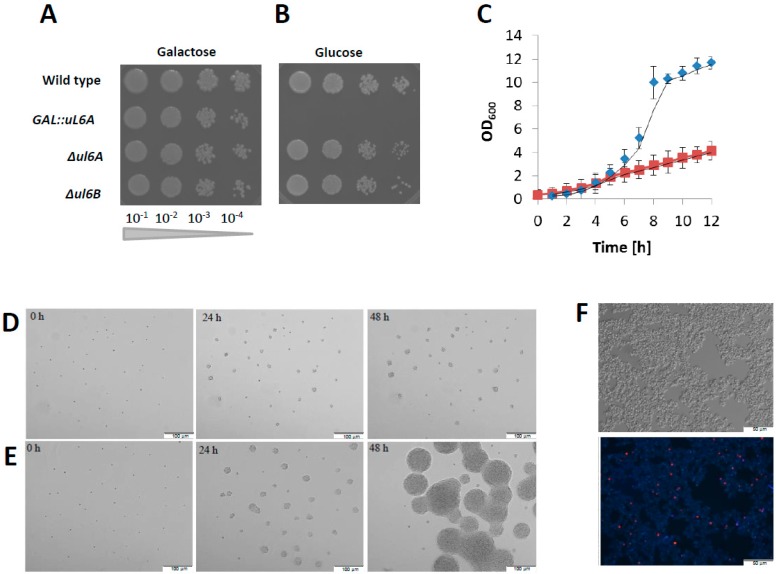
Growth of uL6 mutant yeast strains on various carbon sources. (**A**,**B**) Cells of the *Δul6A, Δul6B*, and GAL::uL6A mutants and the wild type were spotted onto agar plates with YPD (Glucose) or YPGal (Galactose) medium as indicated and incubated at 30 °C for 3 days. (**C**) Growth curves of GAL::uL6A (red squares) and wild type (blue diamonds) strains at 30 °C after shifting exponential cultures from liquid YPGal to liquid YPD medium. The optical density (OD_600_) of the cultures was measured at different time points for up to 12 h. Error bars represent standard deviations obtained from three independent experiments. (**D**–**F**) Effects of depletion of both *uL6A* and *uL6B* on the budding ability and viability. GAL::uL6A mutant cells were grown on YPD (**D**) or YPGal (**E**) plates, and cell growth was monitored under the microscope after 0 h, 24 h, and 72 h. (**F)** Propidium iodide staining of GAL::uL6A budding yeast cells.

**Figure 3 cells-08-00718-f003:**
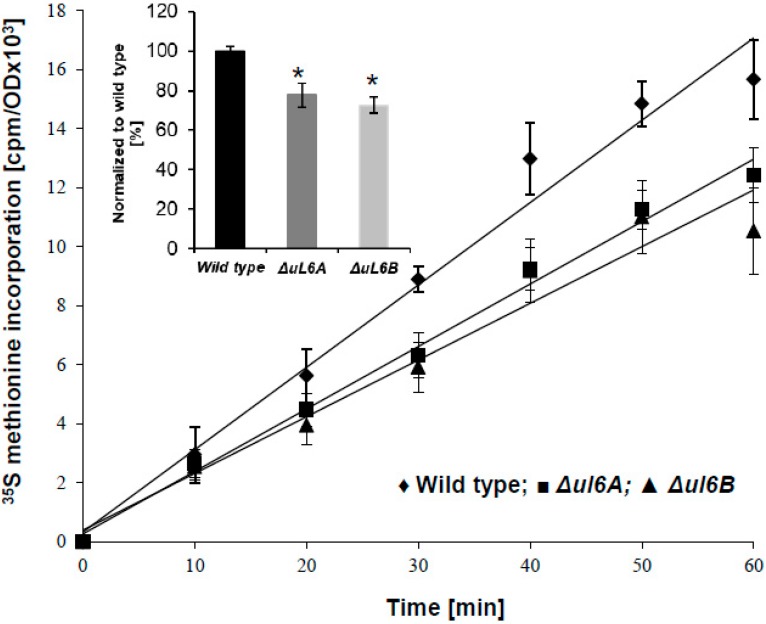
Translational fitness determined as a function of [^35^S]-methionine incorporation. Wild type strain-filled diamonds, *Δul6A* strain-filled squares, and *Δul6B*-filled triangles. Error bars represent standard deviations obtained from three independent experiments. Inset: Translational efficiency normalized to non-treated cells, error bars, SEM (*n* = 3), * *p* < 0.01 Student’s test.

**Figure 4 cells-08-00718-f004:**
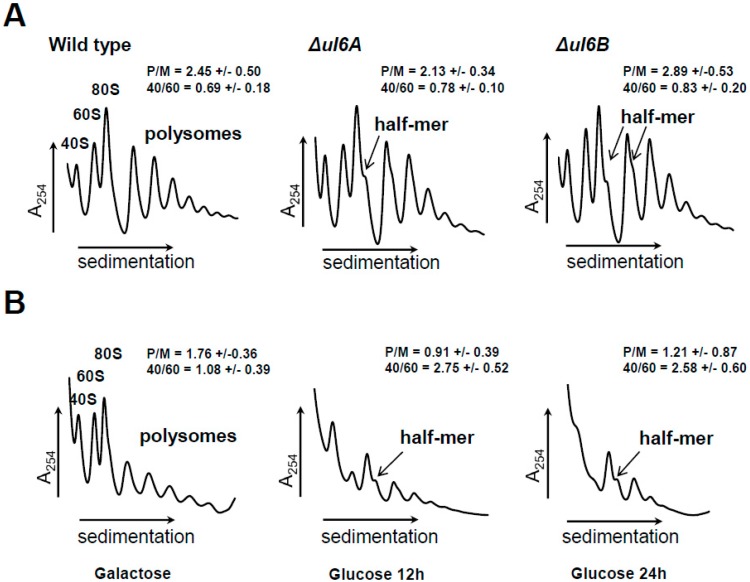
Polysome profile analysis of uL6 mutants. Polysome profile of (**A**) single uL6A or uL6B deletion mutants and wild type strain grown on YPD; (**B**) double uL6A and uL6B mutant grown on YPD for 12 h and 24 h after the shift from the YPGal medium. The sedimentation vector of the ribosomal fractions is indicated by a horizontal arrow, and optical density analysis at 254 nm is shown on the *Y*-axis; the positions of individual ribosomal subunits and half-mers are indicated.

**Figure 5 cells-08-00718-f005:**
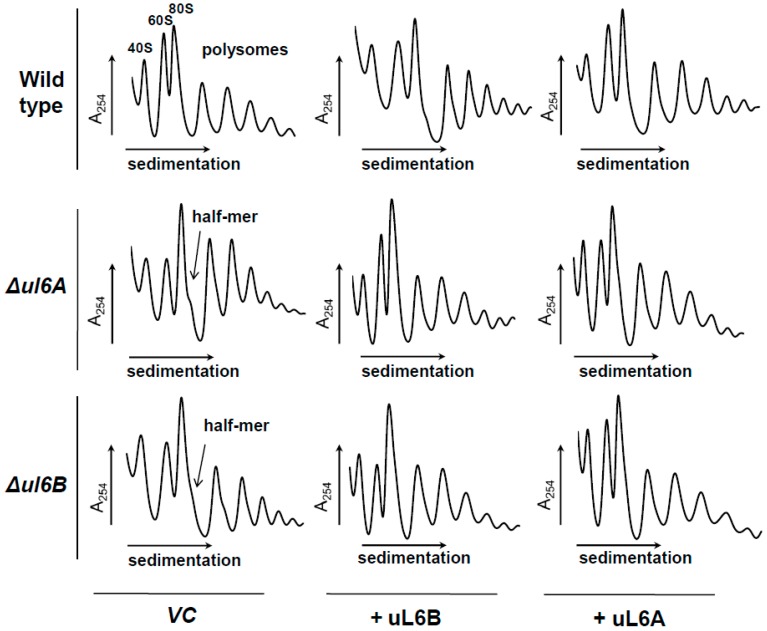
Polysome profile analysis of wild type and single uL6A or uL6B deletion mutants with overexpression of uL6A or uL6B. The yeast strains were grown on SD–Ura medium. The sedimentation vector of the ribosomal fractions is indicated by a horizontal arrow, and optical density analysis at 254 nm is shown on the *Y*-axis; the position of individual ribosomal subunits and half-mers are indicated.

**Figure 6 cells-08-00718-f006:**
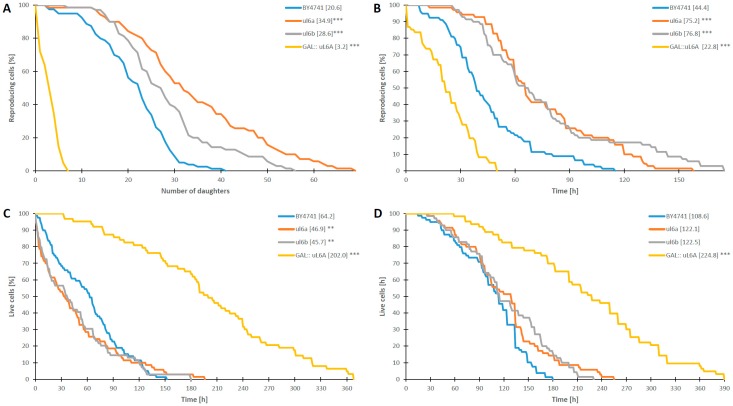
Comparison of the budding lifespan (**A**), reproductive lifespan (**B**), post-reproductive lifespan (**C**), and total lifespan (**D**) of the haploid wild type yeast strain BY4741, isogenic single mutants Δ*ul6A*, Δ*ul6B*, and the GAL::uL6A mutant after cultivation on solid rich YPD media. Statistical significances were assessed using ANOVA and Dunnett’s post hoc test (***p* < 0.05, ****p* < 0.001). Data represent mean values from two independent experiments. The mean value for total 90 cells from two independent experiments is shown in parentheses.

**Figure 7 cells-08-00718-f007:**
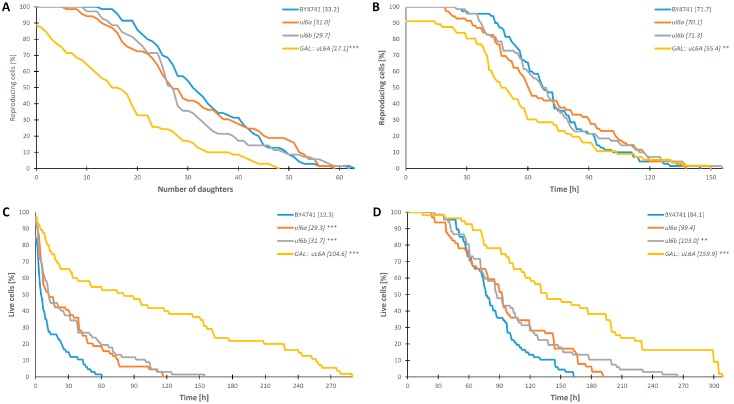
Comparison of the budding lifespan (**A**), reproductive lifespan (**B**), post-reproductive lifespan (**C**) and total lifespan (**D**) of the haploid wild type yeast strain BY4741, isogenic single mutants Δ*ul6A*, Δ*ul6B*, and the GAL::uL6A mutant after cultivation on solid YPGal media. Statistical significances were assessed using ANOVA and Dunnett’s post hoc test (***p* < 0.05, ****p* < 0.001). Data represent mean values from two independent experiments. The mean value for total 90 cells from two independent experiments is shown in parentheses.

**Table 1 cells-08-00718-t001:** Yeast strains used in this study.

Strain Name	Genotype	Source
BY4741	*MAT a; his3Δ1; leu2Δ0; met15Δ0; ura3Δ0*	EUROSCARF
*Δul6A*	*MAT a; his3Δ1; leu2Δ0; met15Δ0; ura3Δ0; YGL147c::kanMX4*	EUROSCARF
*Δul6B*	*MAT a; his3Δ1; leu2Δ0; met15Δ0; ura3Δ0; YNL067w::kanMX4*	EUROSCARF
GAL::uL6A	*MAT a; his3Δ1; leu2Δ0; met15Δ0; ura3Δ0; YGL147c::kanMX4,YNL067w::LEU2 [pYES2-uL6A]*	in this study
BY4741 VC	*MAT a; his3Δ1; leu2Δ0; met15Δ0; ura3Δ0 [pCM190]*	in this study
BY4741 + uL6A	*BY4741; MAT a; his3Δ1; leu2Δ0; met15Δ0; ura3Δ0 [pCM190-uL6A]*	in this study
BY4741 + uL6B	*BY4741; MAT a; his3Δ1; leu2Δ0; met15Δ0; ura3Δ0 [pCM190-uL6B]*	in this study
*Δul6A VC*	*BY4741; MAT a; his3Δ1; leu2Δ0; met15Δ0; ura3Δ0; YGL147c::kanMX4 [pCM190]*	in this study
*Δul6A +* uL6A	*BY4741; MAT a; his3Δ1; leu2Δ0; met15Δ0; ura3Δ0; YGL147c::kanMX4 [pCM190-uL6A]*	in this study
*ΔuL6A +* uL6B	*BY4741; MAT a; his3Δ1; leu2Δ0; met15Δ0; ura3Δ0; YGL147c::kanMX4 [pCM190-uL6B]*	in this study
*ΔuL6B VC*	*BY4741; MAT a; his3Δ1; leu2Δ0; met15Δ0; ura3Δ0; YNL067w::kanMX4 [pCM190]*	in this study
*ΔuL6B +* uL6A	*BY4741; MAT a; his3Δ1; leu2Δ0; met15Δ0; ura3Δ0; YNL067w::kanMX4 [pCM190-uL6A]*	in this study
*ΔuL6B +* uL6B	*BY4741; MAT a; his3Δ1; leu2Δ0; met15Δ0; ura3Δ0; YNL067w::kanMX4 [pCM190-uL6B]*	in this study

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
