# Peer review of "Functional Analysis of the Ribosomal uL6 Protein of Saccharomyces cerevisiae"

_cells, 2019, doi:10.3390/cells8070718_

Round 1

Reviewer 1 Report

In the manuscript by Borkiewicz rt al., the authors compare growth parameters, protein synthesis rates and lifespan of yeast strains with two separate deletions of the paralogs of the ribosomal gene RPL9 (delta uL6a and uL6) and a double mutant in which ul6A is expressed from a galactose – inducible promoter. Single deletions support growth at a rate of 70-80%, whereas the double mutant cannot grow, but mostly retains viability for at least 96 hours. There are expected changes in polysomal profiles, and overexpression of each paralog is capable of compensating the other's deficiency.

Major issues.

1. Deletion of RPL9 in the context of yeast replicative lifespan has been previously done: McCormick et al., A Comprehensive Analysis of Replicative Lifespan in 4,698 Single-Gene Deletion Strains Uncovers Conserved Mechanisms of Aging. Cell Metab. 2015 Nov 3;22(5):895-906. That study must be cited. The authors need to explain if their data is purely confirmatory or if not, in what way it extends those previously published results.

2. In terms of functional differences between the A and B paralogs, they appear functionally identical to me from the data, with the possible differences due to unequal expression levels, which is not analyzed here, but typical for these gene pairs. I assumed the authors held a similar opinion, given that the abstract (line 28) says: “the two uL6 isoforms most likely exhibit the same function”. Then, in the paper's concluding statement (lines 460-61), we read:  “In conclusion, the functional analysis of A and B forms of uL6 protein indicates, that the two uL6 isoforms are not functionally redundant”. This contradicts their own statements like the one in the abstract and pretty much everything they show in terms of the data. What is the reason for having these two opposite statements?

3. The authors should be careful equating change in OD of a culture and increase in cell number. For example, lines 216, 381 - “linear regression” - meaning “linear increase in OD”? The increase in optical density may not reflect the actual increase in cell number, could be a change in cell size, composition, etc.

Minor issues.

415-416  “functional similarity within paralogs is possible due to difference in the level, rate, dynamics or noiseless of expression” - I don't understand the meaning of this phrase.

434 – how exactly is the synthesis of new ribosomes autocatalytic?

436-437 “the genes are duplicated to maintain the assembly homeostasis of translational machinery” - this sounds as if duplication had a purpose; but the authors themselves refer to the global genome duplication event of which the RP genes are some of the remaining survivors.

Table 1 – In the genotype of the GAL::uL6A strain, why is the complementing plasmid shown as [pYES2-RPL6A]? Shouldn't it be RPL9A?

Author Response

Major issues

                1. Deletion of RPL9 in the context of yeast replicative lifespan has been previously done:                 McCormick et al., A Comprehensive Analysis of Replicative Lifespan in 4,698 Single-Gene    Deletion Strains Uncovers Conserved Mechanisms of Aging. Cell Metab. 2015 Nov                3;22(5):895-        906. That study must be cited. The authors need to explain if their data is purely                 confirmatory or    if not, in what way it extends those previously published results.

Yes, the citation by McCormick et al., represents relevant publication and during editing of the manuscript the publication has just escaped our attention, however we were aware of that research. Thus, we have included the publication by McCormic et al. in current version. We would like to add that our data are not purely confirmatory in respect to life-span analysis presented by McCormic et al., who analyzed only reproductive potential (so called replicative lifespan), representing classical older approach, showing only number of daughter cells produced by single mother cell. In turn, here we extended life-span analyses, showing the time of life of single cells during replicative lifespan, which belongs to new approach in analyses of yeast aging. In the presented research, we have shown not only replicative lifespan (what confirms previous data by McCormic et. al. indeed), but more importantly we have provided thorough analysis of total lifespan in time scale, including time during reproduction and time after reproduction (post-reproduction lifespan). We would like to add that the life span analyses presented in the manuscript are performed in broad context of translational machinery functioning, and we think that significantly extend our understanding about modus operandi of two uL6 isoforms.

                2. In terms of functional differences between the A and B paralogs, they appear functionally identical          to me from the data, with the possible differences due to unequal expression levels, which is not   analyzed here, but typical for these gene pairs. I assumed the authors held a similar opinion, given that    the abstract (line 28) says: “the two uL6 isoforms most likely exhibit the same function”. Then, in the                 paper's concluding statement (lines 460-61), we read:  “In conclusion, the functional analysis of A and B     forms of uL6 protein indicates, that the two uL6 isoforms are not functionally redundant”. This            contradicts their own statements like the one in the abstract and pretty much everything they show in      terms of the data. What is the reason for having these two opposite statements?

                We are grateful for this comment. The expression “not functionally redundant“ is of course a mistake in the text. We have corrected the sentence deleting word “not”. Additionally, the presently submitted version of the manuscript was approved by professional English editing service.

                 3. The authors should be careful equating change in OD of a culture and increase in cell number. For           example, lines 216, 381 - “linear regression” - meaning “linear increase in OD”? The increase in optical       density may not reflect the actual increase in cell number, could be a change in cell size, composition,   etc.

                Yes, we agree with the Reviewer's comment, that the increase in optical density might be a result of several factors like increase of the cell number, change in the cell size, composition etc. or combination of them. Therefore, we have also provided the microscopic analyses of the yeast cells, showing that the behavior of cells is significantly changed on the single cell level. We would like to add, that we have observed the linear increase in the OD600nm of GAL::uL6A mutant grown in liquid YPD medium (after shift from YPGal, Fig. 2C), in comparison with the wild type strain, which displayed so-called logarithmic phase or the exponential phase of growth. Thus, to distinguish these two types of growth between wild-type and GAL::uL6A mutant we have used description as “linear regression”. However, we agree with the Reviewer that the use of this phrase might not be appropriate. According to this suggestion, we have corrected the sentence in "...after shift from permissive galactose-based-medium to repressive glucose-based-medium, the GAL::uL6A strain significantly slowed down the growth, and did not show exponential growth as it is the case for wild-type strain (Fig. 2C). As was shown on Fig. 2D-E cell volume does not change during 24 h of incubation. "

 Minor issues

                415-416  “functional similarity within paralogs is possible due to difference in the level, rate,          dynamics or noiseless of expression” - I don't understand the meaning of this phrase.

In this paragraph (lines 411-459) we have considered the metabolic issue related to energy consumption in the cell which maintains duplicated genes, sharing the same ancestral function, especially yeast cell where majority of genes were duplicated.  One of the possible explanation might be the different level, rate, dynamics, or noisiness of duplicated genes expression under selected conditions. However, the Reviewer comment indicates that the sentence the line 414-417 doesn’t reflect our viewpoint and might be confusing for the readers. Therefore,  we have corrected our statement, “As proposed for duplicated genes encoding parts of macromolecular complexes, the functionally similar genes are preserved in one cell due to the different level, rate, dynamics or noisiness of its expression under selected conditions”. In the original sentence there is also a spelling mistake “noiseless” instead noisiness, which was also corrected.

                 434 – how exactly is the synthesis of new ribosomes autocatalytic?

                We have used the word 'autocatalytic' because the biogenesis of ribosomes to some point is 'autocatalytic' self assembling process (as it was shown on the basis of assembling of bacterial ribosomal subunit in vitro), however in the case of eukaryotic ribosomes this process is assisted by more than 200 protein factors and big number of small RNA. Thus, in the case of eukaryotic ribosomes word 'autocatalytic' might be unfortunate, and we have removed this from the text.

                436-437 “the genes are duplicated to maintain the assembly homeostasis of translational machinery” -      this sounds as if duplication had a purpose; but the authors themselves refer to the global genome            duplication event of which the RP genes are some of the remaining survivors.

                Indeed, our intention was not to explaining a purpose for duplication, but rather a purpose of maintenance of duplicated genes within the cell. Therefore, we have corrected this text - : “the genes remain duplicated to maintain the assembly homeostasis of translational machinery”.

                Table 1 – In the genotype of the GAL::uL6A strain, why is the complementing plasmid shown as [pYES2-     RPL6A]? Shouldn't it be RPL9A?

                In this study we are using the new r-proteins nomenclature proposed by Ban et al. 2014 (Curr. Opin. Struct. Biol. 2014, 24:165-9.). Therefore, in the genotype of the  GAL::uL6A strain complementing plasmid shouldn’t be pYES2-RPL6A but pYES2-uL6A, what we have corrected in the Table1.

Reviewer 2 Report

Review report for Manuscript ID cells-536739

 In this report, the authors focused on biological function of the duplicated genes for ribosomal protein uL6 (uL6A and uL6B) in Saccharomyces cerevisiae.  By using ∆uL6A, ∆uL6B, and ∆uL6A∆uL6B mutant strains, and also plasmids for complementation of the proteins into the cells, the authors examined cell growth with galactose- or glucose-medium, and investigated the effects on translational activity, polysome profiles, and cell lifespans.  And the authors concluded that both genes for uL6A and uL6B are required when the cell displays high metabolic rate.  The experimental results are clear and convincing.  The results also seem to be important, because L6 protein is one of components of the ribosomal functional center.  I have only minor comments, as follows.

1)    Figure 2C: Growth curve of GAL::uL6A strain (red squares) shows moderate but significant growth (OD600: o to 4) in 12 h.  I think that this growth may be due to L6 protein synthesized before shifting galactose- to glucose-medium.  The authors need to add a short comment about degradation rate of L6 in yeast cells.

2)    Page 8-9, lines 270-272 “this features were even more pronounced after 24 h of growth on glucose, with further decreasement in polysomal fraction and disappearance of 60S (Fig. 4B)”: in Fig. 4B, polysome fraction at Glucose 24 h (P/M=1.21) increases compared with that at Glucose 12h (P/M= 0.91).  The description should be changed.

3)    Page 9, line 273 “— indicating that lack of uL6 protein is crucial for large ribosomal subunit biogenesis”: this phrase is better to change to “— indicating that presence of uL6 protein is crucial for large ribosomal subunit biogenesis”

4)    Page 11, lines 343-344 “---mainly associated with of inhibition----“: This phrase should be corrected.

5)    Page 13, lines 404-407 “However, uL6, contrary to uL11 which interacts with ribosome in independent manner and does not form extensive interaction network with ribosome, is located close to the critical SRL and form broad web of interactions with rRNA [37]”: In E. coli study, it has been known that uL11 and uL10-L12 complex bind to 23S rRNA in mutually cooperative fashion (Beauclerk et al., 1984), and that uL6 directly binds to SRL (Uchiumi et al., 1999).  This part should be expressed carefully taking these points into consideration. 

Author Response

1)    Figure 2C: Growth curve of GAL::uL6A strain (red squares) shows moderate but significant growth        (OD600: o to 4) in 12 h.  I think that this growth may be due to L6 protein synthesized before shifting            galactose- to glucose-medium.  The authors need to add a short comment about degradation rate of L6         in yeast cells.

                The initial increase in growth rate of GAL::uL6A mutant transferred to YPD medium might be a result of presence of uL6 protein synthesized before shifting from YPGal, indeed; and it is also related to high stability of ribosomal particles; thus, we think that the noticeable growth of the GAL::uL6A mutant strain is mainly associated with the fact that ribosomal particles are maintained in the cells for some period of time, as it was shown in Fig. 4B,  after 12 hours of growth in YPD medium 60S, 80S and some polysomes are present, which may support slow growth. According to Reviewer's suggestion we have added explanation. 

                2)    Page 8-9, lines 270-272 “this features were even more pronounced after 24 h of growth on     glucose, with further decreasement in polysomal fraction and disappearance of 60S (Fig. 4B)”: in Fig.          4B, polysome fraction at Glucose 24 h (P/M=1.21) increases compared with that at Glucose 12h (P/M=       0.91).  The description should be changed.

                In our opinion, the P/M values observed after 12 and 24 hours display similarity (0.91 and 1.21, respectively), however the standard deviation for calculation of P/M value after 24 h is high because the profile is  “flattened” and calculation of the P/M value was very difficult. But this “flattening” of the curve indicates the decrease in polysomal fraction, therefore we have adjusted the text, however we emphasized the lack of 60S fraction, “After 12 h of growth, remarkable reduction in polysome profile was observed, accompanied by simultaneous accumulation of 40S and reduction of 60S subunits; these features were also well visible after 24 h of growth on glucose, with further decreasement in polysomal fraction and especially disappearance of 60S (Fig. 4B)” (line 269-272).

                3)    Page 9, line 273 “— indicating that lack of uL6 protein is crucial for large ribosomal subunit   biogenesis”: this phrase is better to change to “— indicating that presence of uL6 protein is crucial for     large ribosomal subunit biogenesis”

We agree with this suggestion. We have corrected that sentence in the text.

                4)    Page 11, lines 343-344 “---mainly associated with of inhibition----“: This phrase should be corrected.

We have corrected that sentence in the text according to Reviewers suggestion.

                5)    Page 13, lines 404-407 “However, uL6, contrary to uL11 which interacts with ribosome in        independent manner and does not form extensive interaction network with ribosome, is located close          to the critical SRL and form broad web of interactions with rRNA [37]”: In E. coli study, it has been                known that uL11 and uL10-L12 complex bind to 23S rRNA in mutually cooperative fashion (Beauclerk et                 al., 1984), and that uL6 directly binds to SRL (Uchiumi et al., 1999).  This part should be expressed                carefully taking these points into consideration.

                Yes, the Reviewer is correct; stability of ribosomal particle and its action relay on allosteric interplay within the structure, especially the GTPase Associated Center (GAC) represent very good example, where mutual interaction among rRNA and ribosomal proteins play important role in biogenesis and  as well as in the GAC action. The reviewer raised interesting point, and we agree that description about uL6 protein and its interaction with the GAC should be carefully treated. In our description we compared interface rRNA-protein domains between uL6 and uL11, however we agree that functional aspect should be also taken into account. Thus, we have modified the text taking into consideration allosteric nature of the ribosome. 

Reviewer 3 Report

In the manuscript entitled “Functional analysis of the ribosomal uL6 protein of Saccharomyces cerevisiae” the Authors investigated the functional impact of the ribosomal protein uL6 on translational apparatus. The results showed that uL6 is not required to cell survival but it is crucial for efficient ribosome assembly. The analysis of the two uL6 isoforms did not reveal any differences between uL6A and uL6B and demonstrated that the presence of duplicated uL6A and uL6B genes allows yeast cell to efficiently produce uL6 r-protein during high demanding metabolic conditions. In addition, the deletion of single uL6 gene significantly extends lifespan only in cells with high metabolic rate.

Altogether these results led the Authors to conclude that the maintenance of two copies of uL6 gene enables the cell to cope with high demands for effective ribosome synthesis.

In general, the manuscript is well written, the experiments well designed, and the conclusions are of interest for the scientific community.

Minor comments:

-       Authors should add significativity results in Fig. 2C about growth curves.

-       Authors should add “Scale bar” for microscopic images.

-     The number of references is too high. Authors should, where possible, replace some papers with a more recent review. 

Author Response

Minor comments:

                Authors should add significativity results in Fig. 2C about growth curves.

                Yes, we have corrected the description for the growth curves, providing the number of biological analyses.

                Authors should add “Scale bar” for microscopic images.

                We have corrected that in Fig. 2D-F.

                The number of references is too high. Authors should, where possible, replace some papers with a more       recent review. 

                If possible, we would like to keep the bibliography in its present form, because in our opinion it provides thorough insight into the literature related to our presented research.

Round 2

Reviewer 1 Report

The manuscript has improved after the revision. This reviewer's concerns have been satisfactorily addressed.